# Genome-Wide Identification and Characterization of the RWP-RK Proteins in *Zanthoxylum armatum*

**DOI:** 10.3390/genes15060665

**Published:** 2024-05-23

**Authors:** Xianzhe Zheng, Yanling Duan, Huifang Zheng, Hao Tang, Liumeng Zheng, Xiaobo Yu

**Affiliations:** Southwest Research Center for Cross Breeding of Special Economic Plants, School of Life Science, Leshan Normal University, Leshan 614000, China; geminizxz@126.com (X.Z.); duanyanling2023@163.com (Y.D.); zhenghuifang220@163.com (H.Z.); tanghao89@aliyun.com (H.T.); fangmeng2019@gmail.com (L.Z.)

**Keywords:** *Zanthoxylum armatum*, RWP-RK, apomixis, gene expression

## Abstract

Apomixis is a common reproductive characteristic of *Zanthoxylum* plants, and *RWP-RKs* are plant-specific transcription factors known to regulate embryonic development. However, the genome-wide analysis and function prediction of *RWP-RK* family genes in *Z. armatum* are unclear. In this study, 36 *ZaRWP-RK* transcription factors were identified in the genome of *Z. armatum*, among which 15 genes belonged to the RKD subfamily and 21 belonged to the NLP subfamily. Duplication events of *ZaRWP-RK* genes were mainly segmental duplication, and synteny analysis revealed a close phylogenetic relationship between *Z. armatum* and *Arabidopsis*. The analysis of cis-elements indicated that *ZaRWP-RK* genes may be involved in the regulation of the embryonic development of *Z. armatum* by responding to plant hormones such as abscisic acid, auxin, and gibberellin. Results of a real-time PCR showed that the expression levels of most *ZaRWP-RK* genes were significantly increased from flowers to young fruits. Protein–protein interaction network analysis further revealed the potential roles of the ZaRWP-RK proteins in apomixis. Collectively, this study is expected to improve our understanding of *ZaRWP-RK* transcription factors and provide a theoretical basis for future investigations into the *ZaRWP-RK* genes and their regulatory mechanisms in the apomixis process of *Z. armatum*.

## 1. Introduction

Gene transcription is activated or repressed by transcription factors, which bind to specific cis-elements of the promoter [1]. The RWP-RK protein family, with a conserved RWP-RK domain, is a class of plant-specific transcription factors that is expressed in green algae, mosses, and vascular plants [2]. It is divided into two subfamilies: NLPs (NIN-like proteins) and RKDs (RWP-RK domain proteins) [3]. Unlike RKDs, all members of the NLP subfamily possess the PB1 (Phox and Bem 1) domain at their C-termini [4]. Two subfamilies of RWP-RK have been known to participate in the modulation of various processes in plants. Members of the RKD subfamily are primarily involved in embryonic development [5]. For example, *AtRKD4* deficiency may lead to the inhibition of zygote elongation and disrupt early cell division patterns [6]. Overexpression of *AtRKD1* and *AtRKD2* was found to induce the formation of egg-like structures [7]. Members of the NLP subfamily are also linked to the nitrogen response [8]. The AtNLP7 receptor directly binds nitrate in intracellular environments [9]. Loss of *NLP2* function might trigger a decrease in nitrogen fixation and nitrogen content in plants [10]. Moreover, evidence suggests that RWP-RK proteins modulate plant responses to abiotic stress, such as heat stress [11].

Apomixis is an asexual mode of reproduction which bypasses the fertilization stage and produces progenies that are the replica of the mother plant [12]. This reproductive characteristic can greatly reduce the production cost of hybrid seeds, and has an important application value in agricultural breeding. It is generally accepted that there are two types of apomixis: sporophytic and gametophytic. The embryo develops directly from a sporophytic cell (e.g., a nucellar cell) in sporophytic apomixis, while gametophytic apomixis begins with an ’apomeiosis’ cell [12]. Currently, several studies have investigated the genetically related controls of apomixis. For instance, specific expression of *BBM1* in egg cells of rice can induce somatic embryogenesis without fertilization [13]. The *PARTHENOGENESIS (PAR)* gene from dandelion encodes a zinc finger protein that induces egg cell division in lettuce and foxtail millet [14,15]. Similarly, the function and underlying mechanism of *RWP-RK* transcription factors in apomixis have been partially reported. For example, the MITE transposon insertion on the promoter of CitRWP plays an important role in the expression of CitRWP and the emergence of polyembryonic traits produced by nucellar embryony [16]. Transgenic sweet oranges that lack CitRKD1 function undergo no somatic embryogenesis [17]. In addition, overexpression of OsRKD3 in black rice induces somatic embryogenesis [18].

*Zanthoxylum armatum* DC., also known as ‘Tengjiao’ or ‘Qinghuajiao’, belongs to the rutaceae family and is widely distributed in Southwest China [19,20]. The fruit of *Z. armatum* can be utilized to prepare food spice, medicine, and oil, highlighting its significant economic value [21]. Several *Zanthoxylum* species, including *Z. armatum*, have sporophytic apomixis represented by nucellar embryony [22]. Recently, the genome of the diploid, triploid, and tetraploid *Z. armatum* have been reported, which provides support for the gene function mining of *Z. armatum* [23,24,25]. For instance, a homologous gene of *CitRWP*, *Zardc07021*, is identified in *Z. armatum*, which may be a key candidate gene for apomixis [23]. Overexpression of *Zardc07021* in *Arabidopsis* leads to the abnormal development of flower organs and a decrease in pollen viability [23]. Moreover, overexpression of *ZbAGL11* (a class D MADS-box transcription factor) in *Arabidopsis* can induce apomixis after emasculation [26]. To date, analysis of the RWP-RK family has only been conducted for a few species such as soybean, tea, and *Brassica napus* [5,27,28]. These studies analyze the function of RWP-RK transcription factors in defense responses and nitrate responses based on genetic information analysis and RNA sequencing. However, genome-wide identification and characterization analysis of RWP-RK transcription factors in *Z. armatum* and their potential functional prediction in apomixis have not yet been fully investigated. Due to the significance of apomixis in plant breeding, it is imperative to characterize the *RWP-RK* genes of *Z. armatum* and analyze its potential function in apomixis.

In this study, the genome-wide identification and analysis of the *ZaRWP-RK* family was firstly performed in *Z. armatum*. A total of 36 *ZaRWP-RK* transcription factors were identified from the *Z. armatum* genome. Chromosomal localization, phylogenetic relationships, physicochemical properties, gene structure, and cis-elements of the *ZaRWP-RK* transcription factors were explored. Furthermore, the expression levels of *ZaRWP-RK* in different tissues and protein–protein interaction predictions were conducted, which showed that some *ZaRWP-RK* transcription factors may regulate the apomixis process. Therefore, this study is expected to supply more insight into *ZaRWP-RK* transcription factors and provides a theoretical basis for further investigations on RWP-RK genes and their regulatory mechanisms in the apomixis process in *Z. armatum*.

## 2. Materials and Methods

### 2.1. Identification and Physicochemical Properties of ZaRWP-RK Genes

The data of the *Z. armatum* protein sequence and genome were obtained from figshare (https://figshare.com/articles/dataset/Genome_Data_of_Chinese_pepper/20217635; accessed on 28 March 2023), as described by Hu et al. [25]. A hidden markov model (HMM) file of the RWP-RK domain (PF02042) was downloaded from the Pfam database (http://pfam-legacy.xfam.org/; accessed on 3 April 2023), and the candidate ZaRWP-RK proteins were obtained by comparing the *Z. armatum* protein sequence using HMMER 3.0 software with a cut-off value of 0.01 [29]. Then, all candidate ZaRWP-RK proteins were examined by NCBI CDD (https://www.ncbi.nlm.nih.gov/cdd/; accessed on 6 April 2023) and proteins with a conserved RWP-RK domain were retained for subsequent analysis. The number of amino acids, molecular weight (Mw), and isoelectric point value (pI) of ZaRWP-RK proteins were assessed and screened using the ExPasy tools (http://web.expasy.org/protparam/; accessed on 6 April 2023). The subcellular localization of ZaRWP-RK proteins was predicted using WoLF PSORT (https://wolfpsort.hgc.jp/; accessed on 6 April 2023).

### 2.2. Chromosomal Location and Gene Duplication Events

The chromosomal location of ZaRWP-RK genes was identified on the *Z. armatum* genome and visualized with TBtools-II software [30]. The gene duplication events of *ZaRWP-RK* genes were analyzed using the Multiple Collinearity Scan toolkit (MCScanX) [31]. We use the MCscanX tool to obtain a collinearity file that records segmental duplication events. The segmental duplications of the *ZaRWP-RK* transcription factors were screened from the collinearity file and visualized using TBtools-II software, where the red lines represent the segmental duplications. The synteny analysis of the *ZaRWP-RK* genes between *Z. armatum* and other species (*Arabidopsis thaliana*, *Oryza sativa*, and *Citrus grandis*) was visualized using TBtools-II software [30].

### 2.3. Phylogenetic Analysis of ZaRWP-RK Genes

The *Arabidopsis* RWP-RK protein sequences were obtained from the TAIR website (http://www.arabidopsis.org/; accessed on 17 April 2023). The rice RWP-RK protein sequences were downloaded from the MSU TIGR database (http://rice.uga.edu/; accessed on 17 April 2023), and *Citrus grandis* WRKY protein sequences were downloaded from the Citrus Pan-genome2breeding Database (http://citrus.hzau.edu.cn/index.php; accessed on 2 June 2023) (Appendix A). Alignment of RWP-RK protein sequences from *Z. armatum, Arabidopsis thaliana*, *Oryza sativa*, and *Citrus grandis* was achieved using the MAFFT multiple-alignment software [32]. Phylogenetic trees were constructed using the Maximum likelihood (ML) method in raxmlGUI 2.0 with 1000 bootstrap replicates [33]. They were also visualized with the online tool EvolView (https://www.evolgenius.info/evolview-v2/; accessed on 3 January 2024).

### 2.4. Conserved Structural Domains, Conserved Motifs, and Cis-Element Analyses of ZaRWP-RK Genes

Conserved structural domains of ZaRWP-RK proteins were identified using the online tool CD-Search (https://www.ncbi.nlm.nih.gov/Structure/bwrpsb/bwrpsb.cgi, https://www.ncbi.nlm.nih.gov/cdd/; accessed on 25 April 2023) [34]. Subsequently, conserved motifs were determined using the online program MEME (https://meme-suite.org/meme/tools/meme, https://www.ncbi.nlm.nih.gov/cdd/; accessed on 25 April 2023) with a maximum of 10 motifs and a range of motif widths from 6 to 50. Promoter regions 2000 bp upstream of the start codon (ATG) were subjected to cis-element analyses using PlantCARE (https://bioinformatics.psb.ugent.be/webtools/plantcare/html/; accessed on 10 January 2024) and PlantPAN 4.0 (http://plantpan.itps.ncku.edu.tw/plantpan4/promoter_analysis.php; accessed on 12 May 2024) [35]. Next, the conserved structural domains, conserved motifs, and cis-element analyses of *ZaRWP-RK* genes were visualized using TBtools-II software [30].

### 2.5. Plant Materials, RNA Extraction, and qRT-PCR Analysis

Different tissues (female flowers, young fruits, stems, and leaves) of *Z. armatum* were collected from Juzi town in Leshan of Sichuan Province, China (29.32° N, 103.43° E). Total RNA samples were extracted using the FastPure^®^ Plant Total RNA Isolation Kit (Polysaccharides and Polyphenolics-rich) (Vazyme, Nanjing, China) according to the manufacturer’s instructions. An amount of 1 µg of total RNA was then used to synthesize cDNA using ABScript III RT Master Mix for qPCR with gDNA Remover (ABclonal, RK20428, Wuhan, China). The cDNA was subjected to qRT-PCR on a QX96M quantitative PCR system (JLM, Chengdu, China). The reaction protocol was as follows: one cycle at 95 °C for 30 s, 40 cycles at 95 °C for 10 s, 60 °C for 30 s, and a melting curve was generated automatically. The relative expression level was calculated using the 2^−∆∆Ct^ values with the *ZaGAPDH* gene as an internal control. The results were analyzed using IBM SPSS Statistics 27 software. The primers for qPCR were designed using primer premier 5.0 (Appendix A). qRT-PCR was carried out with three biological replicates and each replicate sample was taken from different *Z. armatum* plants.

### 2.6. Protein–Protein Interaction Network Prediction of ZaRWP-RK Proteins

The ZaRWP-RK proteins from *Citrus sinensis* that were homologous to the ZaRWP-RK proteins in *Z. armatum* were submitted to the STRING 12 (https://cn.string-db.org/; accessed on 22 January 2024) to construct a protein–protein interaction network.

## 3. Results

### 3.1. Genome-Wide Identification and Physicochemical Properties of ZaRWP-RK Genes

Based on the RWP-RK domain (PF02042) in the Pfam database, 36 *ZaRWP-RK* genes were identified from the *Z. armatum* genome. The *ZaRWP-RK* genes were then named from *ZaRWP-RK1* to *ZaRWP-RK36*, *and* two *ZaRWP-RKs* (*ZaRWP-RK35* and *ZaRWP-RK36*) were not annotated to chromosomes based on the order of their chromosomal locations (Figure 1). Details of the physical and chemical properties of the *ZaRWP-RK* genes are presented in Appendix A. The coding sequences (CDS) of *ZaRWP-RK* genes ranged from 438 bp (*ZaRWP-RK12*) to 3024 bp (*ZaRWP-RK29*). Similarly, *ZaRWP-RK* genes encode amino acids ranging from 145 aa (*ZaRWP-RK12*) to 1007 aa (*ZaRWP-RK29*), with molecular weights between 17.16 and 110.13 kDa. The pI distribution of *ZaRWP-RK* proteins varied from 4.56 (*ZaRWP-RK17*) to 9.57 (*ZaRWP-RK12*). Most of the *ZaRWP-RK* members were predicted to localize in the nucleus (32), and a few in the cytoskeleton (1), plasma membrane (1), cytosol (1), and chloroplast (1).

Next, we analyzed the duplication events of *ZaRWP-RK* genes (Figure 2). Ten pairs of segmental duplications were identified, while no tandem duplicated genes were found, indicating that segmental duplication likely had a substantial impact on the evolutionary trajectory of the *ZaRWP-RK* family. To gain further insights into the evolutionary relationships among the 36 *ZaRWP-RK* genes between *Z. armatum* and other plants, three collinearity maps were constructed using the *RWP-RK* members of *Z. armatum*, *Arabidopsis thaliana*, *Oryza sativa*, and *Citrus grandis* (Figure 3). The number of orthologous gene pairs between *Z. armatum* and *A. thaliana*, *Oryza sativa*, and *Citrus grandis* was 48, 11, and 35, respectively. Among the three species, *ZaRWP-RK genes* are most closely related to *AtRWP-RK genes*.

### 3.2. Phylogenetic Relationship, Conserved Structural Domains, and Motifs Analysis of ZaRWP-RK Proteins

To explore the phylogeny of the RWP-RK proteins, a phylogenetic tree was constructed based on the amino acid sequences from 36 *Z. armatum* ZaRWP-RK proteins, 14 *Arabidopsis* AtRWP-RK proteins, 13 rice OsRWP-RK proteins, and 8 pummelo CgRWP-RK proteins using raxmlGUI 2.0 (Figure 4). The ZaRWP-RK proteins were clustered into two subfamilies (NLP and RKD) using the results of *A. thaliana* as a reference [36]. Among them, the NLP subfamily contained 21 members (ZaNLP1—ZaNLP21), and the RKD subfamily contained 15 members (ZaRKD1—ZaRKD15) (Figure 4, Appendix A).

To determine the relationships among *ZaRWP-RK* family members, the phylogeny, conserved structural domains, and motif information were analyzed (Figure 5). Analysis of the protein-conserved motifs led to the identification of 10 motifs in ZaRWP-RK proteins, ranging from 21 aa to 50 aa in length (Figure 5B,D). Motifs 2–6 and motifs 8–10 were only found in the ZaNLP subfamily members. It appeared that motif 1 and motif 7 were the most important motifs in the *ZaRWP-RK* family members given that it was identified in almost all proteins. In addition, five conserved structural domains were found in the *ZaRWP-RK* family members. RWP-RK domains were present in almost all members of the *ZaRWP-RK* family, with the exception of *ZaNLP5.* The PB1 domain (including the PB1_NLP and PB1 superfamily) was only identified in ZaNLP proteins, suggesting that the PB1 domain at the ZaNLP proteins was highly conserved.

### 3.3. Cis-Element Analysis of ZaRWP-RK Promoters

To investigate the regulatory mechanisms of *ZaRWP-RK genes*, promoter regions 2000 bp upstream of the start codon (ATG) were subjected to cis-element analyses (Figure 6, Appendix A). Based on the results of cis-acting regulatory element analysis, 21 types of hormone-responsive elements were identified and classified into three categories, including abiotic stress, response to phytohormone, and plant growth and development. Among these elements, most of them were phytohormone-responsive elements. Research has demonstrated a close association of abscisic acid (ABA), auxin, and gibberellin with embryonic development. In terms of hormone response, ABRE (58) elements were the most prevalent and were identified as ABA response elements. Additionally, numerous auxin-responsive elements (14 TGA-element and 4 AuxRR-core) and gibberellin-responsive elements (12 P-box, 8 GARE-motif, and 7 TATC-box) were found in some *ZaRWP-RK* genes. Moreover, among the plant growth- and development-related elements, unnamed__1 elements were the most abundant (42, 60k protein binding site) followed by the CCAAT box (16, MYBHv1 binding site). Interestingly, some cis-elements were specific to certain *ZaRWP-RK* genes, such as MSA-like (*ZaRKD2*), DRE (*ZaNLP10* and *ZaRKD10*), HD-Zip 3 (*ZaNLP13*), and Box III (*ZaRKD1*, *ZaRKD11*, and *ZaRKD12*). In addition, further analysis of the transcription factor binding sites on the promoter of *ZaRWP-RK* genes revealed that multiple transcription factor family members, including AP2, bHLH, MYB, and bZIP, may be involved in regulating the expression of *ZaRWP-RK* genes (Figure 7, Appendix A). These results suggested that the *ZaRWP-RK* genes might be regulated by hormones and some transcription factors, thereby modulating embryonic development.

### 3.4. Tissue-Specific Expression of ZaRWP-RK Genes

In the female flowers of *Zanthoxylum* plants, the embryo is directly developed from a nucellus cell adjacent to the embryo sac without fertilization in the female flower, and leads to the formation of fruits [26]. Therefore, we investigated the expression levels of *ZaRWP-RK* genes in female flowers and young fruits by qRT-PCR, and used the expression levels of *ZaRWP-RK* genes in leaves and stems as controls. To investigate the tissue-specific patterns of *ZaRWP-RK* genes, their expression profiles were explored in four different tissues of *Z. armatum* (female flower, young fruit, stem, and leaf) using qRT-PCR (Figure 8). It was observed that most *ZaRWP-RK* genes were highly expressed in young fruits, suggesting that these genes participate in the regulation of apomixis. Among them, *ZaNLP6*, *ZaNLP10*, *ZaNLP18*, and *ZaRKD3* were significantly upregulated in the young fruit, whereas ZaRKD5 was highly expressed in the female flower and young fruit, indicating their potential role in apomixis. In contrast, the *ZaNLP7* expression was significantly lower in the female flower and young fruit than in the stems and leaf, suggesting it might regulate apomixis in *Z. armatum* plants and perform a negative regulatory function.

### 3.5. Correlation Analysis of Apomixis-Related ZaRWP-RK Genes

Previous research has demonstrated that citrus apomixis may be caused by the expression of a gene known as *CitRWP* (Cg4g018970), which encodes a protein with an RWP-PK domain [16]. Similarly, *OsRKD3* can regulate somatic embryogenesis in black rice [18]. According to the results of the phylogenetic tree analysis, four genes (*ZaRKD2*, *ZaRKD8*, *ZaRKD9*, and *ZaRKD15*) in *Z. armatum* were most closely related to *CitRWP*, and three genes (*ZaRKD10*, *ZaRKD13*, and *ZaRKD14*) were most closely related to *OsRKD3* (Figure 4). *Zanthoxylum* is a close relative of *citrus* in the Rutaceae family [37]. To further investigate the functions of these genes, the STRING database was employed to construct a protein–protein interaction network of these ZaRWP-RK proteins based on their CsRWP-RK homologs (Figure 9). Cs_ont_4g023450 (homolog of ZaRKD2, ZaRKD10, and ZaRKD14) and Cs_ont_5g040080 (homolog of ZaRKD8 and ZaRKD9) were directly connected to ATG5, ATG12, AP2A1, AP2S1, CHC, and FIS1. Moreover, Cs_ont_5g040080 (homolog of ZaRKD8 and ZaRKD9) directly interacted with Alba3.

## 4. Discussion

The RWP-RK transcription factor is widely expressed in many plants and regulates various physiological processes such as nitrogen response, gametophyte development, and abiotic stress regulation [38]. In many plants, including *Arabidopsis*, rice, elephant grass, and soybean, members of the RWP-RK transcription factor family have been reported [5,36,39,40]. However, no study has analyzed this family in *Z. armatum*. In this study, we performed the genome-wide identification of the RWP-RK family members in *Z. armatum* and analyzed their expression during reproductive processes, based on the recently published genome information of *Z. armatum* [25].

Despite ongoing research on the karyotype analysis of *Zanthoxylum* plants since 1996, the determination of chromosome number and ploidy has remained elusive due to the complexity of the species [41,42]. Recent advancements in the technology of genome sequencing and assembly have enabled multiple research groups to conduct genome analysis on various *Zanthoxylum* plants in recent years. Luo et al. [43] think that *Z. armatum* had karyotypes of 2n = ~128. *Z. armatum* (2n = 66) genome assembly was performed by Wang et al. [23]. A total of 68 pairs of chromosomes were found in the genome of *Zanthoxylum bungeanum*, as described by Feng et al. [37]. Hu et al. [25] think that *Z. armatum* belongs to the allotetraploid with a karyotype of 2n = 4x = 132. Recently, a triploid *Z. armatum* genome was assembled with a karyotype of 3x = 99 [24]. However, the conclusions regarding the genomic ploidy of *Z. armatum* are still not perfectly resolved. In this study, 36 members of the RWP-RK family were identified according to the tetraploid *Z. armatum* genome, which surpassed that reported in other plants—for example, 14 in *Arabidopsis*, 13 in rice, and 8 in pummelo (Figure 1 and Figure 4)—indicating the significant expansion throughout the evolutionary process, probably due to the diversity of their functions. It is worth noting that 39 *RWP-RK* genes were screened in the genome of diploid *Z. armatum*, as described by Wang et al. [23], which is not half that of the tetraploid *Z. armatum*. It is unclear whether this situation is due to differences in materials or differences in genomic annotation. Further elucidation of chromosome ploidy will deepen our understanding of the evolution of the RWP-RK family in *Z. armatum.*

Gene duplication is a significant driving force for the diversification of gene functions and species evolution [44]. In eukaryotes, there are two main types of duplication events: tandem and segmental [45]. Previous studies have shown that the NLP subfamily in *Arabidopsis* evolved through segmental duplication rather than tandem amplification [36]. In this study, we used the genomic information of the tetraploid *Z. armatum* to identify 10 pairs of segmental duplications, but only 5 pairs of *ZaRWP-RK* genes were located on different chromosomes of the same subgenome, indicating that these genes may have undergone segmental duplication events, and the rest may have originated from different subgenomes (Figure 2). It is worth noting that gene density is significantly higher in the telomeres of the chromosomes, and some *ZaRWP-RK* genes are located in the ends of chromosomes, inferring the potential roles of *ZaRWP-RK* genes in telomeres (Figure 1 and Figure 2). Moreover, we observed that the number of orthologous gene pairs between *Z. armatum* and dicots (*Arabidopsis* and pummelo) exceeded that between *Z. armatum* and monocots (rice), indicating that after the divergence of monocots and dicots, the *ZaRWP-RK* transcription factors have undergone extensive evolution and duplication (Figure 3).

The RWP-RK transcription factor family is generally divided into two subfamilies: the RKD subfamily and the NLP subfamily [36]. The available studies indicate that in *Arabidopsis*, the RKD subfamily has five members and the NLP subfamily has seven members [39,46]. In rice, there are seven members in the RKD subfamily and six members in the NLP subfamily [36]. In this study, we analyzed 36 *ZaRWP-RK* transcription factors from the *Z. armatum* genome. Through a phylogenetic analysis, we classified 15 of these members into the RKD subfamily and 21 members into the NLP subfamily (Figure 4, Appendix A). Notably, the transcriptional activity, DNA-binding activity, and protein–protein interactions of transcription factors are often controlled by their motifs [47]. Among the 10 conserved motifs identified in *ZaRWP-RK* family, motifs 1 and 7 were expressed in all *ZaRWP-RK* members, indicating that they might constitute the most conserved part of ZaRWP-RK proteins (Figure 5B). Motifs 2–6 and 8–10 were only identified in members of the NLP subfamily, suggesting that these motifs might play an important role in unique biological functions of the NLP subfamily. Analysis of the conserved domains of ZaRWP-RK proteins showed that the PB1 domain was present in all NLP subfamily members, which is the main distinguishing point between the RKD and NLP subfamilies (Figure 5C). Interestingly, our analyses showed that ZaNLP2 contained a conserved GAF_2 domain. A recent study reported that nitrate-triggered ROS signaling and the detection of nitrate deficiency are dependent upon the GAF domain of AtNLP7 [48]. Therefore, we speculated that ZaNLP2 might be involved in the regulation of nitrogen response. In addition, the PoIC superfamily domain was detected in ZaRKD10. PoIC is a major DNA polymerase involved in genome duplication, suggesting that ZaRKD10 might participate in the rapid cell division stage, such as in embryonic development [49].

The activation of gene expression is regulated by cis-elements present in the promoter region [50]. Therefore, the physiological functions of cis-elements on the promoter can be used to predict the biological function of the genes [51]. In the *ZaRWP-RK* family, 21 types of cis-elements were identified, categorized into abiotic stress (6), response to phytohormone (9), and plant growth and development (6) (Figure 6, Appendix A). Most of them were hormone response elements, with the ABRE (ABA response element) accounting for the largest proportion. ABAs are key hormones that control plant growth, embryogenesis, seed physiology reactions, and stress tolerance [52]. Pretreatment with ABAs in the embryo maturation medium increased the cassava somatic embryo conversion into plants [53]. In addition, a significant number of auxin response elements and gibberellin response elements were identified. Auxin and gibberellin have been shown to regulate plant embryo development [54,55]. Furthermore, numerous elements related to plant growth and development were identified in the promoters of *ZaRWP-RK* members, suggesting that *ZaRWP-RK* transcription factors participate in embryogenesis. Consistent with this view, numerous studies have shown that *ZaRWP-RK* modulates plant reproduction processes [7]. Among the five RKD members in *Arabidopsis*, *AtRKD1* and *AtRKD4* are highly expressed during early embryo development, and the *AtRKD4* mutant leads to abnormal embryo development [6,39]. The MpRKD controls the development of gametophytes and contributes to the formation of gemma cups [56]. Therefore, our results indicate that *ZaRWP-RK* transcription factors may participate in plant reproductive processes by responding to and regulating hormones such as ABA, auxin, and gibberellin.

Apomixis, found in over 300 genera and 40 families throughout the plant kingdom, displays a wide taxonomic distribution [12]. In this mechanism, the fertilization step is skipped, allowing cells from nucellus or integument tissues to develop into somatic embryos. Previous studies have highlighted the crucial role of RWP-RK transcription factors in both somatic embryogenesis and apomixis processes [17,18,23]. Therefore, investigating the expression pattern of the *ZaRWP-RK* transcription factors during the apomictic reproduction process in *Z. armatum* is of great importance. In this study, we found that the expression levels of most *ZaRWP-RK* genes increase from flowers to young fruits significantly (Figure 8). Among them, the expression levels of *ZaNLP6*, *ZaNLP10*, *ZaNLP18*, *ZaRKD3*, and *ZaRKD5* show the most significant differences, indicating their potential functions in apomixis. In contrast, *ZaNLP7* expression was significantly lower in the female flower and young fruit than in the stems and leaf, suggesting that it might negatively regulate apomixis.

Autophagy proteins are required for multiple functions during embryogenesis [57]. Our predictions based on the protein–protein interaction network suggested that Cs_ont_4g023450 (homolog of ZaRKD2, ZaRKD10, and ZaRKD14) and Cs_ont_5g040080 (homolog of ZaRKD8 and ZaRKD9) may interact with ATG5 and ATG12, suggesting that ZaRKDs may be involved in autophagy-regulated embryonic development (Figure 9). In addition, our results showed that Cs_ont_4g023450 and Cs_ont_5g040080 may interact with AP2A1, AP2S1, and CHC1 (Figure 9). The adaptor protein complex 2 (AP-2) proteins participate in various biological processes, such as plant growth and development, and stresses response [58]. AP2σ and AP1/2β modulate embryogenesis and plant growth in *Arabidopsis* [59,60]. AP-2 has been shown to interact with CHC and regulate flower organ development. Therefore, we speculate that ZaRKDs may interact with AP2A1, AP2S1, and CHC1, thereby influencing the reproductive process in *Z. armatum*. Moreover, we also found that Cs_ont_4g023450 and Cs_ont_5g040080 interacted with MIS1 and Alba3. The mitochondrial fission complex has been implicated in *Arabidopsis* heat tolerance [61]. ALBA proteins modulate various stress responses [62]. Recent studies also have shown that overexpression of one *RWP-RK* gene in pearl millet can significantly enhance plant heat tolerance [11]. Therefore, it can be hypothesized that ZaRKDs may regulate heat tolerance in *Z. armatum*.

## 5. Conclusions

In this study, we identified 36 *ZaRWP-RK* transcription factors in the genome of *Z. armatum*. Most *ZaRWP-RK* genes were distributed on 24 chromosomes, except *ZaRWP-RK35* and *ZaRWP-RK36*. The *ZaRWP-RK* genes were divided into two subfamilies, NLP and RKD, through phylogenetic analysis. The conserved structural domain and motif analysis revealed the functional similarity and specificity of these *ZaRWP-RK* genes. Meanwhile, cis-element analysis showed that the *ZaRWP-RK* genes had multiple functions in plant reproductive processes. Analysis of gene expression in different tissues and the protein–protein interaction network further highlighted the potential regulatory role of *ZaRWP-RK* genes in the apomixis of *Z. armatum*. In summary, through genome-wide identification and characterization of the RWP-RK transcription factor in *Z. armatum*, our results not only improve our understanding of the *ZaRWP-RK* family but also offer theoretical support for further studies into the regulatory mechanism of apomixis in *Z. armatum*.

## Figures and Tables

**Figure 1 genes-15-00665-f001:**
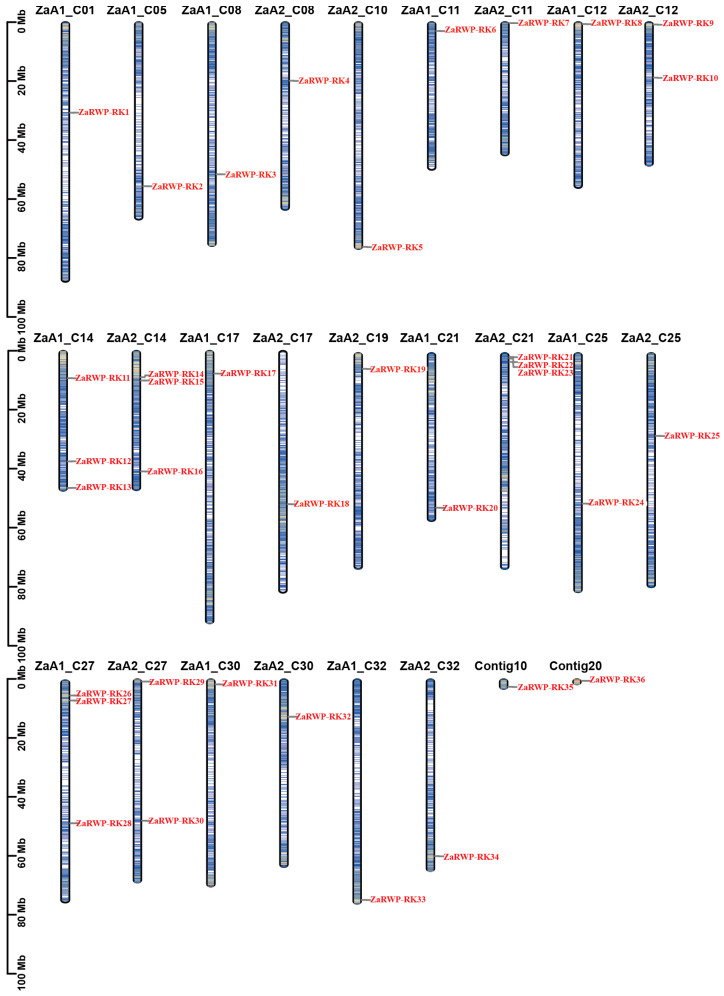
Chromosomal locations of the *ZaRWP-RK* genes. The chromosomal information of the *ZaRWP-RK* genes was available in the database of the *Z. armatum* genome. ZaA1 and ZaA2 represent the allelic chromosomes of autotetraploid. *ZaRWP-RK* genes are numbered in the order of chromosomes.

**Figure 2 genes-15-00665-f002:**
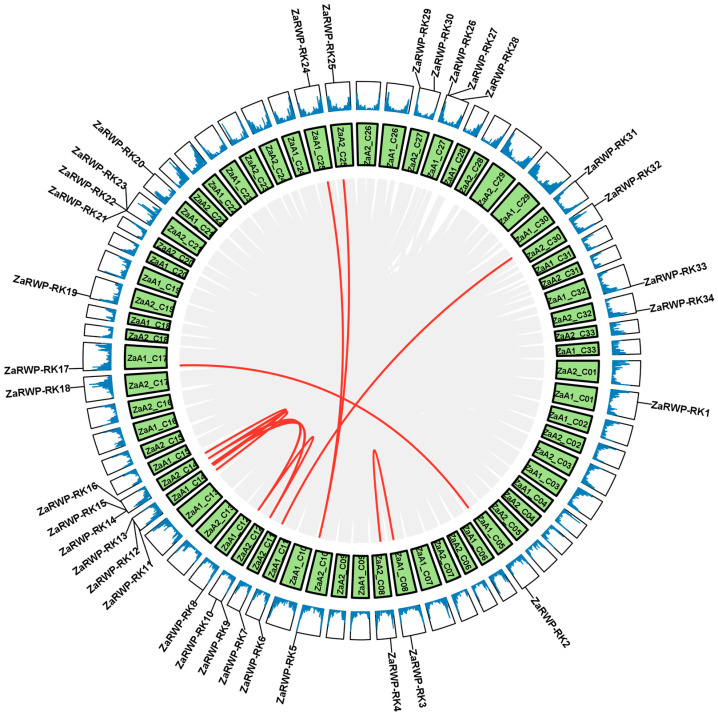
Collinearity analysis of the *ZaRWP-RK* genes. The gray lines inside the circle indicate all the collinear gene pairs within the *Z. armatum* chromosomes. Red lines represent segmental duplication events in *ZaRWP-RK* family. The texts in the green boxes indicate the chromosome numbers. Gene density is shown by the blue lines in the outermost box, and higher values indicate higher densities. All *ZaRWP-RK* genes have their location on the chromosomes indicated by a black line.

**Figure 3 genes-15-00665-f003:**
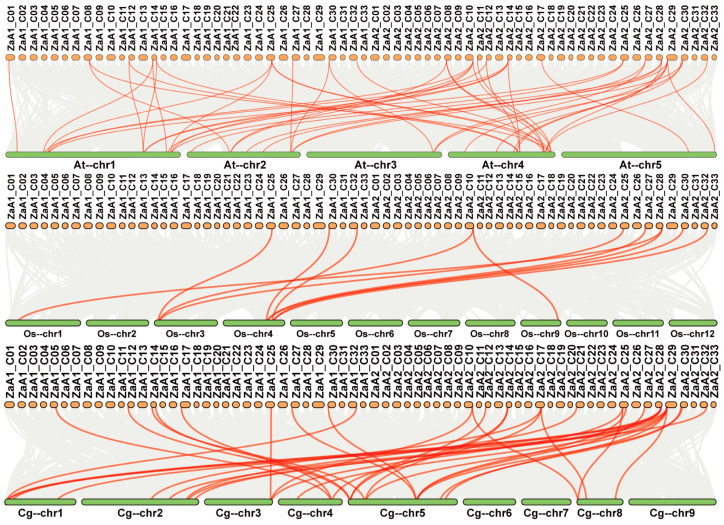
Synteny analysis of *ZaRWP-RK* genes between *Z. armatum*, *Arabidopsis thaliana*, *Oryza sativa*, and *Citrus grandis*. Za, *Z. armatum.* At, *Arabidopsis thaliana*. Os, *Oryza sativa.* Cg, *Citrus grandis.* The gray lines in the backdrop indicate all the collinear gene pairs between *Z. armatum* and three other species, while the red lines signify the collinear pairs of *ZaRWP-RK* genes.

**Figure 4 genes-15-00665-f004:**
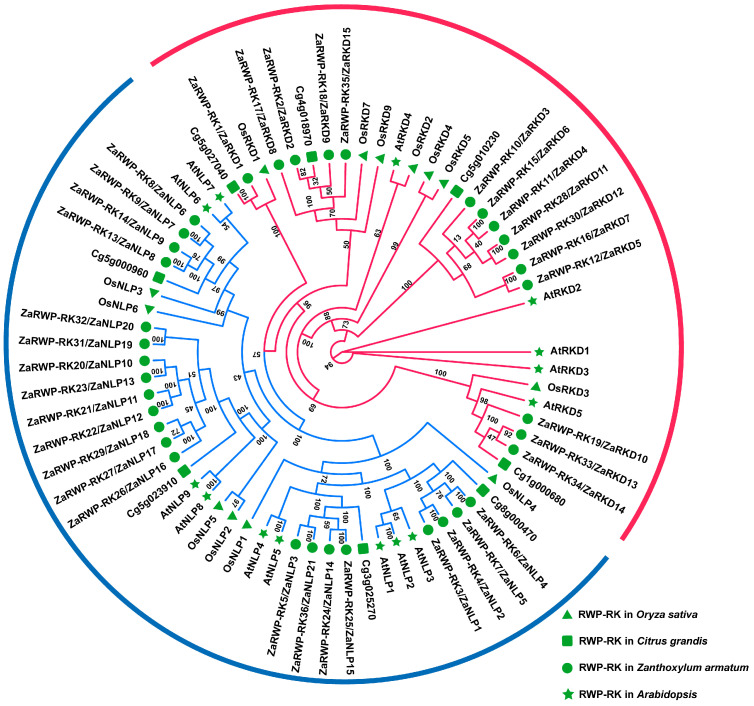
Phylogenetic relationships between ZaRWP-RK proteins and other plant RWP-RK proteins. The phylogenetic trees were constructed using the Maximum likelihood (ML) method in raxmlGUI 2.0 with 1000 bootstrap replicates and displayed using EvolView. The red lines represent the RKD subfamily, and blue lines represent the NLP subfamily. The green markers of different shapes represent the four species.

**Figure 5 genes-15-00665-f005:**
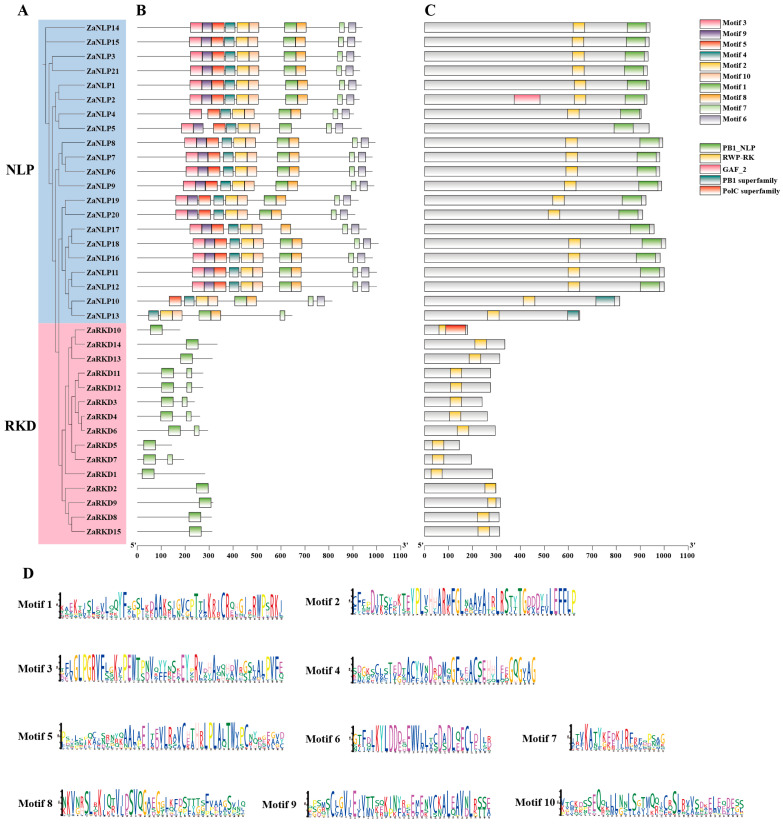
Phylogenetic relationships, conserved motifs, and conserved structural domains of ZaRWP-RK proteins in *Z. armatum*. (**A**) Phylogenetic tree of 36 ZaRWP-RK proteins. (**B**) Visualized location of conserved motifs in ZaRWP-RK proteins. Each of the ten motifs is distinguished by a different color. (**C**) Visualized location of conserved structural domains in ZaRWP-RK proteins. Different domains are presented in different colored boxes. (**D**) Sequences logos of the 10 conserved motifs in ZaRWP-RK proteins.

**Figure 6 genes-15-00665-f006:**
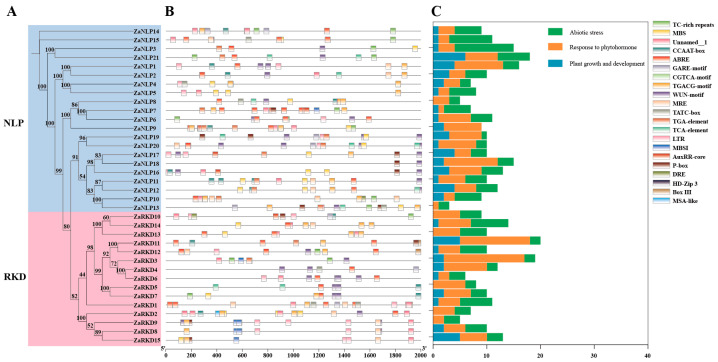
Analysis of cis-elements in the promoter of the *ZaRWP-RK* genes. (**A**) Phylogenetic tree of 36 ZaRWP-RK proteins. (**B**) Visualized location of cis-elements in the promoter of the *ZaRWP-RK* genes. Each of the cis-elements is distinguished by a different color. (**C**) The number of cis-elements for each *ZaRWP-RK* gene in three categories.

**Figure 7 genes-15-00665-f007:**
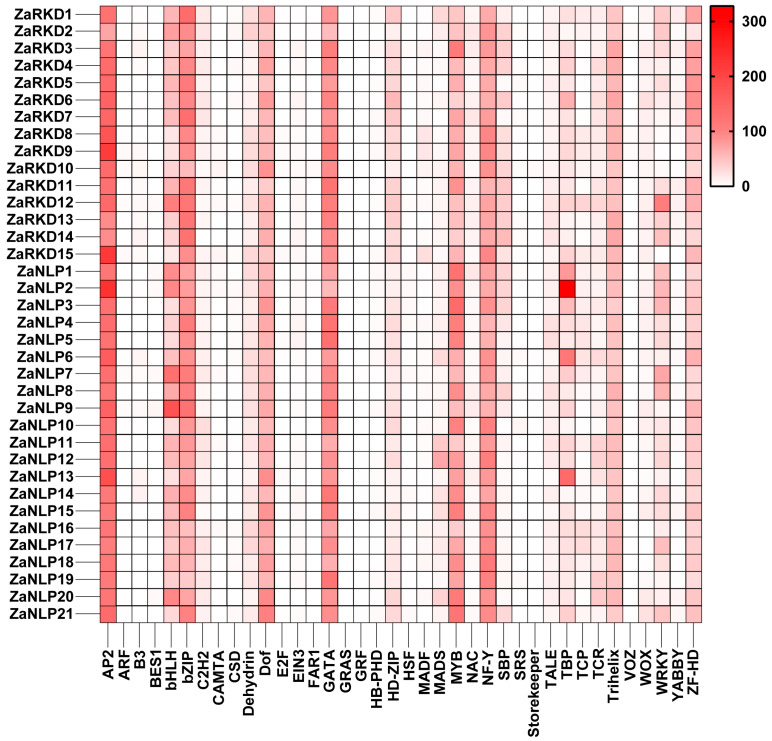
Statistics for transcription factor binding sites in the promoter regions of *ZaRWP-RK* genes. Colors with darker tints have more binding sites.

**Figure 8 genes-15-00665-f008:**
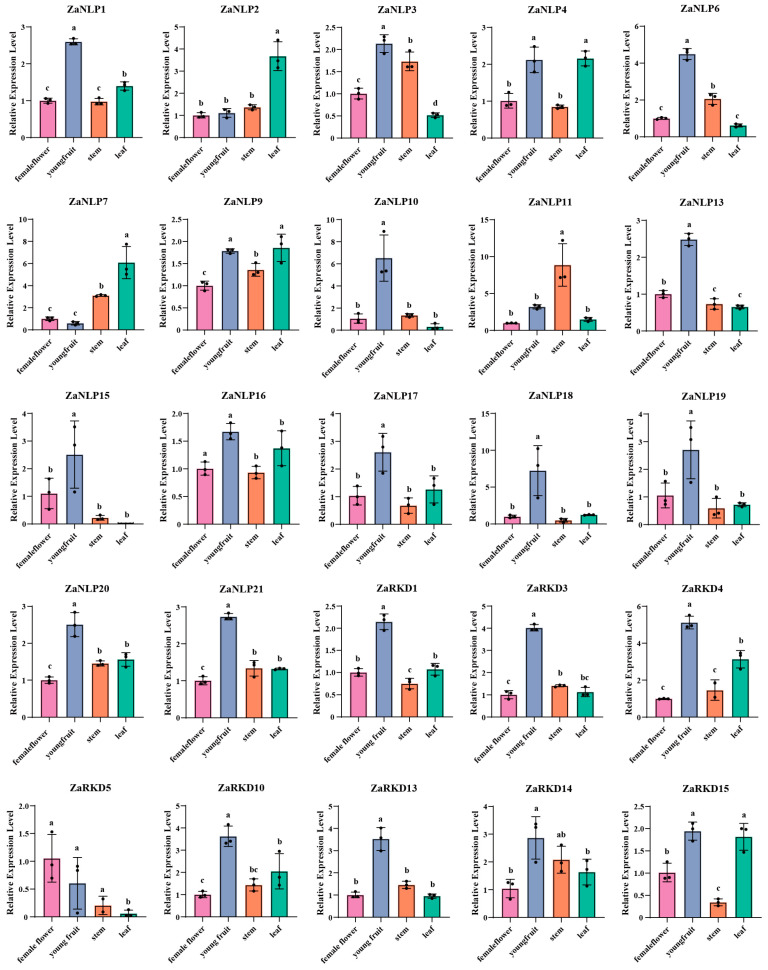
Expression patterns of *ZaRWP-RK* genes in different tissues of *Z. armatum*. Data are presented as the mean ± SD of three independent experiments. Different letters indicate significant differences between different tissues (*p* < 0.05), as determined by Student’s *t*-test.

**Figure 9 genes-15-00665-f009:**
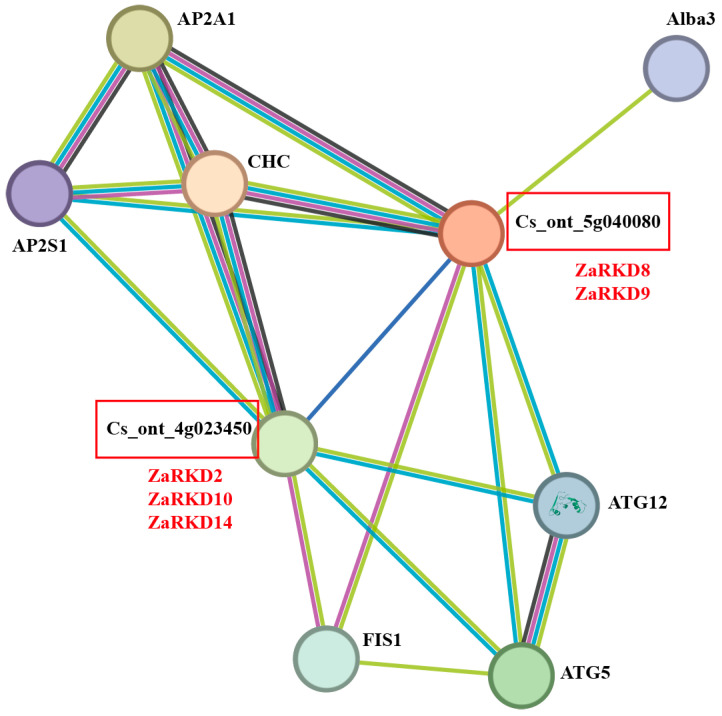
Protein–protein interaction network of RWP-RK homologs in *Citrus sinensis* visualized by STRING.

## Data Availability

The main data supporting the results of this article are included within this article (and its Appendix A).

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
