# Peer review of "Genome-Wide Identification and Characterization of the RWP-RK Proteins in *Zanthoxylum armatum"

_genes, 2024, doi:10.3390/genes15060665_

Round 1

Reviewer 1 Report

Comments and Suggestions for Authors

The manuscript discussing the genome-wide identification and characterization of the RWP-RK transcription factor family in Zanthoxylum armatum is undeniably important and offers potential insights into apomixis regulation. However, the manuscript as it stands is substantially lacking in several key areas, necessitating significant revisions for it to meet publication standards. The current introduction does not meet expectations for a clear and impactful beginning to this work which is interesting. The absence of a clearly articulated research objective or hypothesis significantly undermines the reader's ability to grasp the fundamental purpose of the study. This section needs a complete overhaul by clearly stating the main purpose of work and it would benefit the science community. Mthods-1. Ortholog Analysis: The description of methods used for identifying orthologs is vague and insufficient. Specific details regarding the tools and databases utilized is needed to complete this section. Without this clarity, the scientific validity of the entire study is called into question.

2. Motif Analysis:The explanation of motif analysis is critically incomplete. It is essential to specify the genomic context in relation to the start codon to ensure that the findings are interpreted correctly and are scientifically valid.

3. Software Considerations:The use of outdated software such as PlantCARE for predicting cis-regulatory elements is concerning, despite this is a useful tool, please ensure that the its updated because from my experiences they dont keep this up to date. However, please  justify for its use or consider of more current tools for relevance and accuracy.

4. Quantitative PCR (qPCR) Protocols:The lack of detail regarding the replication of qPCR experiments is a major flaw. Replicates -- how many was used and please state this is  fundamental to establishing the reliability and statistical validity of experimental data. Additionally, the rationale behind the selection of specific tissues for study is missing, I was unsure if this was mention as to why they were preferred over the other tissue sets,  this is  critical oversight  and should be addressed to justify the experimental design. The figures, particularly Figure 2, are of poor quality and lack necessary explanatory captions. The presence of an unexplained purple confused  me. Lastly, for the supplementary data it would be helpful to outlined the ortholog information as a table, this  not present. 

Comments on the Quality of English Language

Overall the quality of  English is not an issue in this paper but other technical issues need to be address. However, the writers could use an English speaker to help with the overall flow of the paper. 

Author Response

Dear Reviewer,

Thanks very much for taking your time to review this manuscript. I really appreciate all your comments and suggestions! Please find my itemized responses in below and my revisions/corrections in the re-submitted manuscript.

Thanks again!

1. Comment: The current introduction does not meet expectations for a clear and impactful beginning to this work which is interesting. The absence of a clearly articulated research objective or hypothesis significantly undermines the reader's ability to grasp the fundamental purpose of the study. This section needs a complete overhaul by clearly stating the main purpose of work and it would benefit the science community.

1. Reply:We gratefully appreciate for your thoughtful comment. The aim of our study was to perform a genome-wide identification and characterization of RWP-RK transcription factors in armatum and explore their potential functions in apomixis. Indeed, we lack a description of the background and purpose of the study in the Introduction section. We've carefully revised the Introduction section. Thanks again for your valuable comment.

2. Comment: Ortholog Analysis: The description of methods used for identifying orthologs is vague and insufficient. Specific details regarding the tools and databases utilized is needed to complete this section. Without this clarity, the scientific validity of the entire study is called into question.

2. Reply:We gratefully appreciate for your thoughtful comment. We have supplemented and revised our methods used for identifying orthologs. Thanks again for your valuable comment.

3. Comment: Motif Analysis:The explanation of motif analysis is critically incomplete. It is essential to specify the genomic context in relation to the start codon to ensure that the findings are interpreted correctly and are scientifically valid.

3. Reply:We gratefully appreciate for your thoughtful comment. We have supplemented and revised our methods used for motif analysis. Thanks again for your valuable comment.

4. Comment: Software Considerations:The use of outdated software such as PlantCARE for predicting cis-regulatory elements is concerning, despite this is a useful tool, please ensure that the its updated because from my experiences they dont keep this up to date. However, please justify for its use or consider of more current tools for relevance and accuracy.

4. Reply:We gratefully appreciate for your thoughtful comment. Initially, we used PlantCARE for cis-element analysis, based on other research. Although the tool is useful, we do overlook the timeliness of the tool. Therefore, we performed a supplemental analysis of cis-elements using the recently updated tool, PlantPAN 4.0. The results are presented in Figure 7 and Supplementary Table 2. Thanks again for your valuable comment.

5. Comment: Quantitative PCR (qPCR) Protocols:The lack of detail regarding the replication of qPCR experiments is a major flaw. Replicates -- how many was used and please state this is  fundamental to establishing the reliability and statistical validity of experimental data.

5. Reply:We gratefully appreciate for your thoughtful comment. The relative expression level was calculated using the 2−∆∆Ct values with ZaGAPDH gene as internal control. qRT-PCR was carried out with three biological replicates and each replicate sample was taken from different Z. armatum plants. We've supplemented this in the Methods section. Thanks again for your valuable comment.

6. Comment:Additionally, the rationale behind the selection of specific tissues for study is missing, I was unsure if this was mention as to why they were preferred over the other tissue sets, this is critical oversight and should be addressed to justify the experimental design.

6. Reply:We gratefully appreciate for your thoughtful comment. In the female flowers of Zanthoxylum plants, the embryo is directly developed from a nucellus cell adjacent to the embryo sac without fertilization in female flower, and leads to the formation of fruits. Therefore, we investigated the expression levels of ZaRWP-RK genes in female flowers and young fruits by qRT-PCR, and used the expression levels of ZaRWP-RK genes in leaves and stems as controls. We've supplemented this in the Results section. Thanks again for your valuable comment.

7. Comment:The figures, particularly Figure 2, are of poor quality and lack necessary explanatory captions. The presence of an unexplained purple confused me.

7. Reply:We gratefully appreciate for your thoughtful comment. We apologize that the purple dot was added by mistake when we organized the image and does not represent any results. We have revised and improved the resolution of Figure 2. Thanks again for your valuable comment.

8. Comment:Lastly, for the supplementary data it would be helpful to outlined the ortholog information as a table, this not present.

8. Reply:We gratefully appreciate for your thoughtful comment. Indeed, we left out homologous gene information from other species used in this study. We have added ortholog information in Supplementary Table 2. Thanks again for your valuable comment.

Reviewer 2 Report

Comments and Suggestions for Authors

Dear Editor, 

The work is excellent, the authors presented and discussed the results in a clear, concise, and in-depth way. I will only have two points as suggestions:

- make the objective of the work clear in the abstract and introduction;

- need to review the last paragraph of the introduction, since the authors present a summary of the conclusion, without addressing the hypothesis and objective of the work;

- check the possibility of enlarging the letter in the figures.

Author Response

Dear Reviewer,

Thanks very much for taking your time to review this manuscript. I really appreciate all your comments and suggestions! Please find my itemized responses in below and my revisions/corrections in the re-submitted manuscript.

Thanks again!

1. Comment: make the objective of the work clear in the abstract and introduction.

1. Reply: We gratefully appreciate for your thoughtful comment.The aim of our study was to perform a genome-wide identification and characterization of RWP-RK transcription factors in Z.armatumand explore their potential functions in apomixis. Indeed, we lack a description of the background and purpose of the study in the Introduction section. We've carefully revised the Abstract and Introduction section. Thanks again for your valuable comment.

2. Comment: need to review the last paragraph of the introduction, since the authors present a summary of the conclusion, without addressing the hypothesis and objective of the work.

2. Reply: We gratefully appreciate for your thoughtful comment.We've carefully revised the Introduction section. Thanks again for your valuable comment.

3. Comment: check the possibility of enlarging the letter in the figures.

3. Reply: We gratefully appreciate for your thoughtful comment.We've improved the clarity of the Figures. Thanks again for your valuable comment.

Reviewer 3 Report

Comments and Suggestions for Authors

I have some remarks:

Line 8-10. The text is unclear please check the grammar, there are some errors e. g “

 Zanthoxylum armatum is an forest tree with apomixis characteristic”. Please check the grammar in the text to ensure that it is error-free.

Line 38. Do you mean loss of function of NLP2?

Line 44. Please explain the meaning of the term "apomixis", along with its biological and economic significance.

Line 119. Please review the use of abbreviations, you previously defined the isoelectric point abbreviation.

Line 129, Could you please provide additional details on how you locate the segmental duplications?

Figure 2. Why do only certain outermost boxes have legends? Indicate that they are the ZaRWP-RK genes.

Figure 3. T The grey lines in the backdrop that indicate collinear connections between species do not provide clear information. Everything looks grey.

Figure 4. The resolution of the image needs to be improved, as the bootstrap value is not distinguishable.

Line 206. Indicate for the qPCR analysis the internal control housekeeping genes used and the method used to validate expression.

Line 264. Please unify the naming convention for the species. Should it be written as "Arabidopsis" or "Arabidopsis" in italics?

Comments on the Quality of English Language

 Moderate editing of English language is required

Author Response

Dear Reviewer,

Thanks very much for taking your time to review this manuscript. I really appreciate all your comments and suggestions! Please find my itemized responses in below and my revisions/corrections in the re-submitted manuscript.

Thanks again!

1. Comment: Line 8-10. The text is unclear please check the grammar, there are some errors e. g“Zanthoxylum armatum is an forest tree with apomixis characteristic”. Please check the grammar in the text to ensure that it is error-free.

1. Reply: We gratefully appreciate for your thoughtful comment. We have rewritten this sentence as follows: “Apomixis is a common reproductive characteristic of Zanthoxylumplants, and RWP-RKs are plant-specific transcription factors known to regulate embryonic development.” Thanks again for your valuable comment.

2. Line 38. Do you mean loss of function of NLP2?

2. Reply: Yes, what I'm trying to mean here is the lack of NLP2 functionality.We have rewritten this sentence as follows: “Loss of NLP2 function might trigger a decreased in nitrogen fixation and nitrogen content in plants.” Thanks again for your valuable comment.

3. Comment: Line 44. Please explain the meaning of the term "apomixis", along with its biological and economic significance.

3. Reply: Apomixis is an asexual mode of reproduction which bypasses the fertilization stage and produce progenies that are the replica of the mother plant.This reproductive characteristic can greatly reduce the production cost of hybrid seeds, and has important application value in agricultural breeding.We've added the above to the Introduction section. Thanks again for your valuable comment.

4. Line 119. Please review the use of abbreviations, you previously defined the isoelectric point abbreviation.

4. Reply: We gratefully appreciate for your thoughtful comment.Indeed, we have defined the isoelectric point abbreviation in Materials and Methods section, and repeating the definition here. We have rewritten this sentence as follows: “The pI distribution of ZaRWP-RK proteins varied from 4.56 (ZaRWP-RK17) to 9.57 (ZaRWP-RK12).” Thanks again for your valuable comment.

5. Line 129, Could you please provide additional details on how you locate the segmental duplications?

5. Reply: We gratefully appreciate for your thoughtful comment.We use the MCscanX tool to obtain a collinearity file that records segmental duplications events. The segmental duplications of the ZaRWP-RK transcription factors were screened from the collinearity file and visualized using TBtools-II software, where the red lines represent the segmental duplications (Figure 2). We've added the above to the Materials and Methods section. Thanks again for your valuable comment.

6. Figure 2. Why do only certain outermost boxes have legends? Indicate that they are the ZaRWP-RK genes.

6. Reply: We gratefully appreciate for your thoughtful comment. In Figure 2, the outermost boxes also represent the chromosomes, and gene density is shown by the blue lines in the outermost box. ZaRWP-RKgenes are located in different chromosomesand pointed with a black line. We have improved the resolution of Figure 2 to make it clearer. Thanks again for your valuable comment.

7. Figure 3. The grey lines in the backdrop that indicate collinear connections between species do not provide clear information. Everything looks grey.

7. Reply: We gratefully appreciate for your thoughtful comment. In Figure 3, the gray lines represent all the colinear gene pairs between the genomes of two species, so it will look many. Only the collinear relationships of the ZaRWP-RK transcription factorsare identified with red lines. We have revised the legend of Figure 3. Thanks again for your valuable comment.

8. Figure 4. The resolution of the image needs to be improved, as the bootstrap value is not distinguishable.

8. Reply: We gratefully appreciate for your thoughtful comment.We have improved the resolution of Figure 4. Thanks again for your valuable comment.

9. Line 206. Indicate for the qPCR analysis the internal control housekeeping genes used and the method used to validate expression.

9. Reply: We gratefully appreciate for your thoughtful comment.The relative expression level was calculated using the 2−∆∆Ctvalues with ZaGAPDH gene as internal control. We've added the above to the Materials and Methods section. Thanks again for your valuable comment.

10. Line 264. Please unify the naming convention for the species. Should it be written as "Arabidopsis" or "Arabidopsis" in italics?

10. Reply: We gratefully appreciate for your thoughtful comment.We have uniformly named ”Arabidopsis” in italics. Thanks again for your valuable comment.

Round 2

Reviewer 1 Report

Comments and Suggestions for Authors

The introductions much better! 

Comments on the Quality of English Language

Please read over and check for English language.